# Bacterial Communities in the Rhizosphere at Different Growth Stages of Maize Cultivated in Soil Under Conventional and Conservation Agricultural Practices

Yendi E. Navarro-Noya,[a] Yosef Chávez-Romero,[b] Stephanie Hereira-Pacheco,[b] Arit Seleny de León Lorenzana,[b] Bram Govaerts,[c,d] Nele Verhulst,[c] Luc Dendooven[b]

aLaboratory of Biotic Interactions, Centro de Investigación en Ciencias Biológicas, Universidad Autónoma de Tlaxcala, Tlaxcala, Mexico
bLaboratory of Soil Ecology, Cinvestav, Mexico City, México
cInternational Maize and Wheat Improvement Centre (CIMMYT), El Batán, Texcoco, México
dCornell University, Ithaca, New York, USA

**ABSTRACT** Farmers in Mexico till soil intensively, remove crop residues for fodder and grow maize often in monoculture. Conservation agriculture (CA), including minimal tillage, crop residue retention and crop diversification, is proposed as a more sustainable alternative. In this study, we determined the effect of agricultural practices and the developing maize rhizosphere on soil bacterial communities. Bulk and maize (*Zea mays* L.) rhizosphere soil under conventional practices (CP) and CA were sampled during the vegetative, flowering and grain filling stage, and 16S rRNA metabarcoding was used to assess bacterial diversity and community structure. The functional diversity was inferred from the bacterial taxa using PICRUSt. Conservation agriculture positively affected taxonomic and functional diversity compared to CP. The agricultural practice was the most important factor in defining the structure of bacterial communities, even more so than rhizosphere and plant growth stage. The rhizosphere enriched fast growing copiotrophic bacteria, such as Rhizobiales, Sphingomonadales, Xanthomonadales, and Burkholderiales, while in the bulk soil of CP other copiotrophs were enriched, e.g., *Halomonas* and *Bacillus*. The bacterial community in the maize bulk soil resembled each other more than in the rhizosphere of CA and CP. The bacterial community structure, and taxonomic and functional diversity in the maize rhizosphere changed with maize development and the differences between the bulk soil and the rhizosphere were more accentuated when the plant aged. Although agricultural practices did not alter the effect of the rhizosphere on the soil bacterial communities in the flowering and grain filling stage, they did in the vegetative stage.

**IMPORTANCE** We studied the effect of sustainable conservation agricultural practices versus intensive conventional ones on the soil microbial diversity, potential functionality, and community assembly in rhizosphere of maize cultivated in a semiarid environment. We found that conservation agriculture practices increased the diversity of soil microbial species and functions and strongly affected how they were structured compared to conventional practices. Microbes affected by the roots of maize, the rhizobiome, were different and more diverse than in the surrounding soil and their diversity increased when the plant grew. The agricultural practices affected the maize rhizobiome only in the early stages of growth, but this might have an important impact on the development of maize plant.

**KEYWORDS** community assembly, functional diversity, intensive agricultural practices, tillage, sustainable agriculture, plant microbiome

Address correspondence to Yendi E. Navarro-Noya, nyendi@hotmail.com.

The authors declare no conflict of interest.

The bacterial community structure in soil is determined by different interacting factors, such as climate, soil characteristics and vegetation (1). Organic material is used as C substrate by heterotrophic microorganisms (mostly archaea, bacteria and

fungi) and the environment in which they are active controls their metabolic activity (2). In an agroecosystem, soil microbial communities are also defined by the intensity of agricultural practices and the cultivated crop (3–5) where the mutualistic interactions between the crop and soil microorganisms in the rhizosphere play a crucial role in the productivity of the system (6). Soil microorganisms in the rhizosphere are affected by the plant through root secretions and other rhizodeposits, as plants can secrete between 20 and 40% of their photosynthates in the rhizosphere (7, 8). As such, the microbial biomass in the rhizosphere is higher although generally less diverse than in the surrounding bulk soil (9). Conversely, the stimulated microorganisms in the rhizosphere actively contribute to plant development through nutrients released after organic matter mineralization (6).

Maize (*Zea mays* L.) is a globally important crop that is used for a variety of food and industrial products (10). Since the beginning of the 1990s, the "International Maize and Wheat Improvement Center (CIMMYT)" has promoted conservation agriculture (CA), which aims to increase soil organic matter content and water availability, reverse soil fertility loss, and reduce management costs and soil erosion (11, 12) while maintaining or increasing crop yields. Conservation agriculture includes crop diversification, minimum tillage and crop residue retention to replace more conventional practices (CP) that rely on intensive tillage, removal of crop residue and monoculture (13). The effect of CA versus CP on yield and soil properties has been investigated at different experimental sites, such as '*El Batán*' in a semiarid environment in Mexico (e.g., [14, 15]). Crop yields, soil fertility and soil organic matter content increased when CA was applied compared to CP, while other soil characteristics, such as pH, electrolytic conductivity (EC) and particle size distribution, were similar.

In a previous study of the soil bacterial community structure in CP and CA, it was found on the one hand that the relative abundance of phylotypes belonging to bacterial groups that preferred low nutrient environments was higher in soil with CA (e.g., Acidobacteria, Planctomycetes, and Verrucomicrobia) compared to CP (16). On the other hand, the relative abundance of phylotypes belonging to bacterial groups that preferred nutrient rich environments, such as Actinobacteria, showed an opposite trend. We hypothesized that the bacterial communities in the rhizosphere of maize plants cultivated in CA and CP would resemble each other more than in the bulk soil of both treatments as i) the available organic material in the rhizosphere will be less dependent on agricultural practices than in the bulk soil and ii) the plant host more than soil characteristics defines the root microbiome assemblages (17). As such, we predict that the species turnover in the rhizosphere is lower than in the bulk soil.

The age or stage of development of the plant affects the composition of the plant microbiome (18). The age-related assembly of the microbiome may be associated with root growth, physiology, architecture, morphology, and exudates. The composition of the root exudates and the microorganisms in the rhizosphere change along plant growth (19) and some evidence suggests that plants can obtain the necessary nutrients during their different growth stages through the stimulation of their root and rhizosphere microbiome (20, 21). Wattenburger et al. (22) found the greatest shifts in bacterial community composition in the rhizoplane of maize during the vegetative stage and that crop diversification increased the abundance of Verrucomicrobia and Acidobacteria in the rhizoplane during this period. We hypothesize that (i) the species and functional diversity are highest during the later stages of plant development when the root structure is large and complex, and (ii) agricultural practices (CA versus CP) affect microbial communities most during early stages when the plant establishes itself. To test these hypotheses, the bacterial community composition was determined in the bulk soil and the rhizosphere of maize cultivated in soil under CA and CP during the vegetative, flowering and grain filling stage of the plants. The functional profile of the bacteriome was obtained with the "Phylogenetic Investigation of Communities by Reconstruction of Unobserved States" software (PICRUSt). The bacterial alpha and beta, and functional diversity was determined.

## RESULTS

A total of 72 soil samples were collected (*N* = 72), i.e., two agricultural practices (CP

**TABLE 1** Characteristics of soil at the experimental site in El Batán (Mexico)

| Treatment | pH | EC[a] (dS m$^{-1}$) | (g kg$^{-1}$ soil) | | | | | | USDA textural classification |
|---|---|---|---|---|---|---|---|---|---|
| | | | Total N | Organic C | WHC[b] | Clay | Sand | Silt | |
| CP[c] | 6.0 A[d] | 0.59 A | 1.01 A | 29.0 B | 627 A | 320 A | 430 A | 250 A | Clay loam |
| CA[e] | 6.1 A | 0.69 A | 0.96 A | 38.4 A | 633 A | 330 A | 430 A | 240 A | Clay loam |

[a]EC: Electrolytic conductivity.
[b]WHC: Water holding capacity.
[c]CP: Conventional practices: crop residue removed, maize monoculture (*Zea mays* L.) and conventional tillage since 1991.
[d]Values with the same capital letter are similar for the different agricultural practices, i.e., within the column.
[e]CA: Conservation agriculture: crop rotation of maize and wheat (*Triticum aestivum* L.), residue retention and zero tillage since 1991.

and CA) × two different soil samples (bulk and rhizosphere soil) × collected three times during maize plant development (vegetative, flowering and grain filling stage) × from two plots and × three maize plants in each plot, and the physicochemical characteristics were determined (see Materials and Methods, Fig. S1). The organic C content was significantly higher in CA (38.4 ± 4.1 g kg$^{-1}$) than in the CP soil (29.0 ± 0.4 g kg$^{-1}$; $P < 0.05$), but the other soil characteristics were similar (Table 1).

Metabarcoding analysis was done with the V3-V4 regions of the 16S rRNA genes and sequenced with 300-PE runs in an Illumina MiSeq. Sequencing retrieved a total of 4,324,922 high quality sequences representing 4,546 amplicon sequence variants (ASVs). The four most abundant phyla were Proteobacteria (relative abundance 57.6 ± 15.3%), Acidobacteria (13.8 ± 5.4%), Actinobacteria (7.4 ± 2.9%), and Bacteroidetes (4.3 ± 2.3%) (Fig. 1). The four most abundant genera were *Halomonas* (25.1 ± 17.4%), *Pseudomonas* (2.3 ± 3.2%), *DA101* (1.3 ± 0.8%), and *Bacillus* (0.6 ± 0.2%).

**Effect of agricultural practices on soil bacterial communities.** True taxonomic and functional diversity were calculated as equivalent Hill numbers, i.e., effective number of ASVs, at $q$ diversity orders of 0, 1 and 2 which consider all ($q = 0$), frequent ($q = 1$) and dominant ASVs ($q = 2$). The effective number of ASVs (true taxonomic diversity) and the mean functional diversity (true functional diversity) was significantly higher in CA than in CP at all $q$ diversity levels ($P < 0.05$) (Fig. 2A and B). A compositional approach was used to investigate the bacterial community composition and analyze its structure (see Materials and

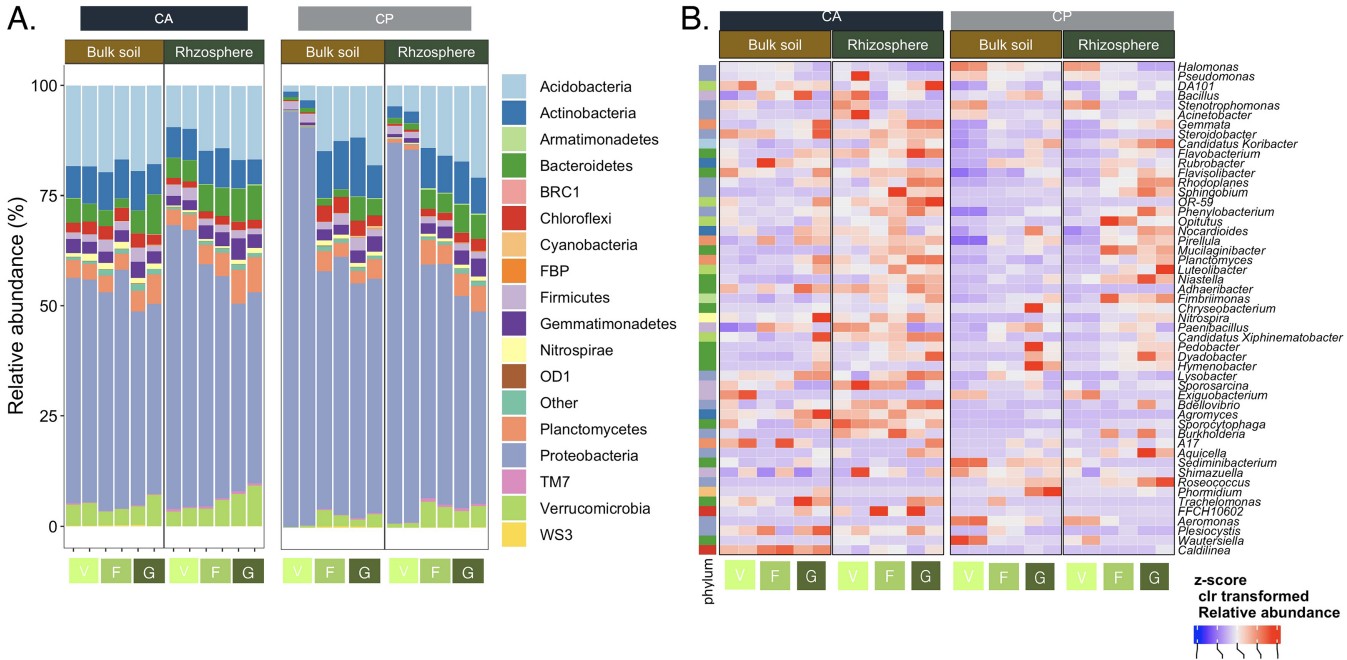

**FIG 1** Taxonomic distribution of the bacterial communities in the rhizosphere of maize (*Zea mays* L.) sampled in the vegetative (V), flowering (F) and grain filling stage (G) and bulk soil under conservation agriculture (CA) or conventional practices (CP). A) Bar-plot with the relative abundance of bacterial phyla, and B) heat-map with the z-score of the centered log-ratio (*clr*) transformed frequencies of the 50 most abundant bacterial genera.

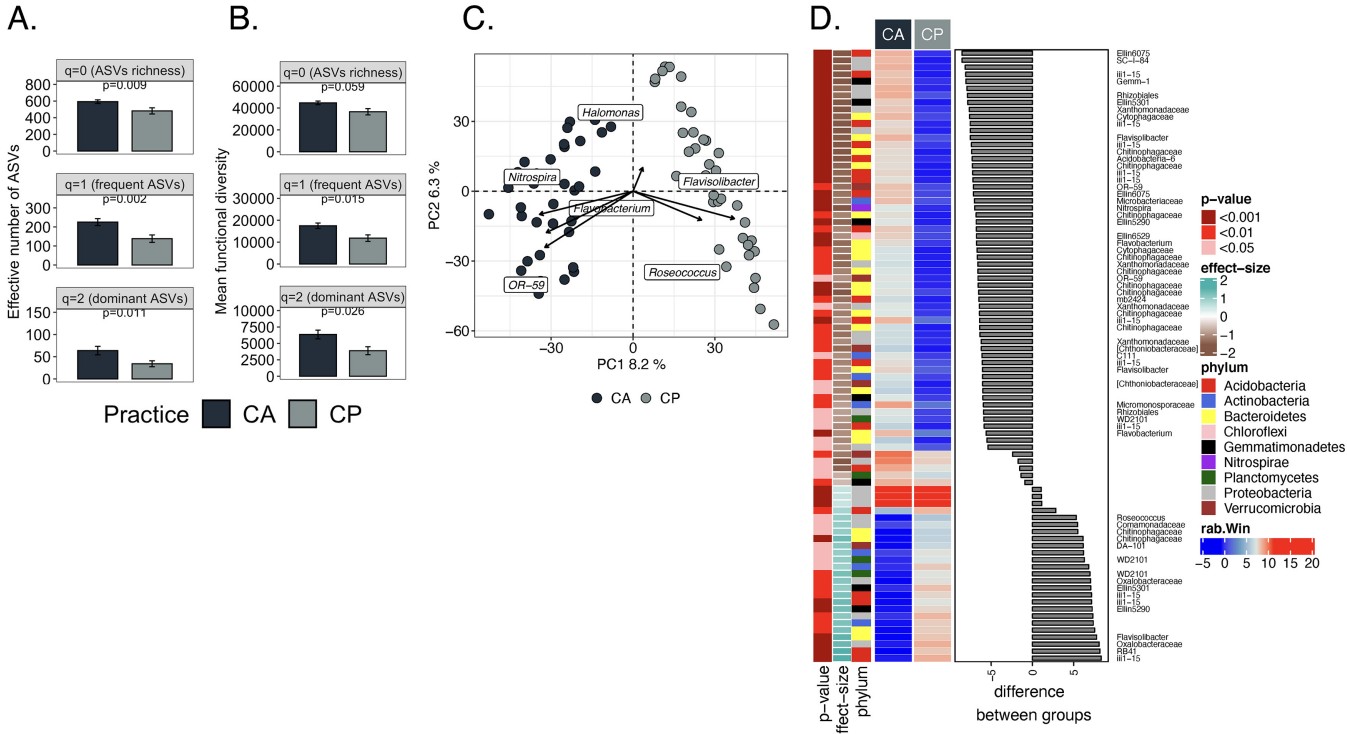

**FIG 2** Effect of agricultural practices, i.e., conservation agriculture (CA) and conventional practices (CP), on soil bacterial communities. A) Hill numbers of alpha taxonomic diversity, B) Hill numbers of mean functional diversity. The bar height indicates the mean for each agricultural practice and the error line shows the standard error of the mean. Linear mixed effects models with 1000 Monte-Carlo permutations and plot in the field as random factor was used to test the significant differences (*P*-value). C) Principal-component analysis (PCA) with all amplicon sequence variants (ASVs). D) Differentially abundant ASVs between CA and CP as determined by an ANOVA-Like Differential Expression tool for compositional data and Benjamini-Hochberg sequential correction. Median centered log-ratio (*clr*) transformed frequencies (rab.win), *P*-values and the effect size were plotted as heat-map and the median difference between species as bar-plot.

Methods; [23]). The principal-component analysis (PCA) clearly separated the bacterial communities in CA from those in CP (Fig. 2C). The permuted multivariate analysis of variance (perMANOVA) with Aitchison distances to test the effect of the agricultural practices indicated a large and highly significant effect on the bacterial community that explained 45% of the total variance ($P < 0.001$, Table 2).

The effect of different treatments on the ASVs and enzyme classification numbers (EC numbers) were determined with an ANOVA-like differential expression (ALDEx; [24]). A total of 87 ASVs was affected significantly by agricultural practices, most of them belonging to Acidobacteria, Bacteroidetes, and Proteobacteria (Fig. 2D). Sixty-two ASVs were enriched in CA and they belonged mostly to the families Chitinophagaceae, Rhizobiales, and Xanthomonadaceae, while CP enriched 25 ASVs belonging mostly to the genera *Halomonas*, *Roseococcus*, and *Flavisolibacter*.

The frequency of 187 EC numbers (functions) was affected significantly by agricultural practices (CA versus CP) of which 41 were more abundant under CA and 146

**TABLE 2** Permutated analysis of variance (perMANOVA) to test the effect of treatment[a]

| Effect | Df | Sum of squares | R² | F | P |
|---|---|---|---|---|---|
| Treatment | 1 | 70.3 | 0.452 | 298.7 | 0.001 |
| Soil sample | 1 | 32.7 | 0.210 | 138.8 | 0.001 |
| Maize development stage | 2 | 24.8 | 0.159 | 52.6 | 0.001 |
| Treatment × maize development stage | 2 | 12.3 | 0.079 | 26.1 | 0.001 |
| Soil sample × maize development stage | 2 | 1.5 | 0.010 | 3.3 | 0.019 |

[a]perMANOVA analysis with the Aitchison distances. Conservation agriculture (CA); conventional practices (CP); soil, bulk or rhizosphere soil; development stage, vegetative, flowering and grain filling stage of maize (*Zea mays* L.) on the soil bacterial community structure.

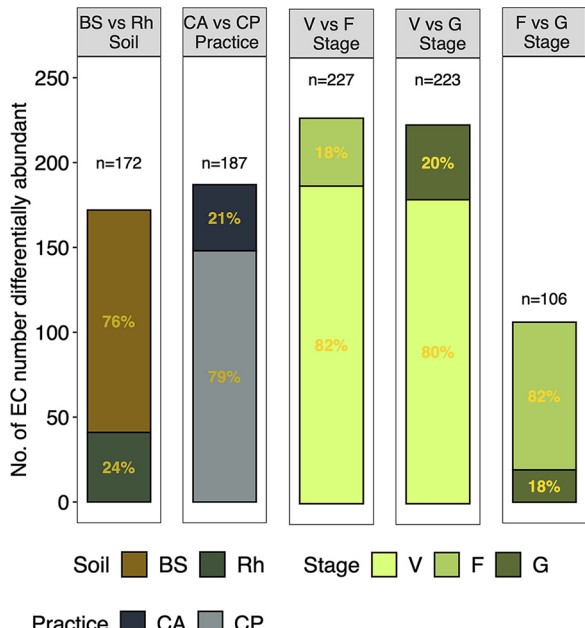

**FIG 3** Number of differentially abundant enzyme classification numbers (EC numbers) as determined by an ANOVA-Like Differential Expression tool for compositional data and Benjamini-Hochberg sequential correction. Effect of agricultural practices, i.e., conservation agriculture (CA) and conventional practices (CP), of maize (*Zea mays* L.) rhizosphere (Rh) versus the bulk soil (BS) and the maize rhizosphere in the vegetative (V), flowering (F) and grain filling (G) growth stages on soil bacterial communities.

under CP (Fig. 3). The functions more abundant under CA were related to biosynthesis and degradation/utilization/assimilation processes in equal proportions (Fig. S2). The predicted functions involved in biosynthesis and to a lesser extend in degradation/utilization/assimilation, and generation of precursor metabolites and energy were more abundant under CP. Notably, the function with the highest positive effect size due to CA was methanogenesis from acetate (effect size = 0.90, $P < 0.001$).

**Effect of the maize rhizosphere on the soil bacterial communities.** The species richness, expressed as effective number of ASVs at diversity order $q = 0$, was similar in the rhizosphere of maize and in the bulk soil (Fig. 4A). Meanwhile, the effective number of frequent ($q = 1$) and dominant ($q = 2$) ASVs was significantly higher in the rhizosphere than in the bulk soil ($P < 0.01$). The functional diversity at orders $q = 1$ and 2 was also significantly higher in the maize rhizosphere than in the bulk soil ($P < 0.001$; Fig. 4B).

The rhizosphere had a highly significant effect on the bacterial community structure considering ASVs and the effect explained 21% of the total variance ($P < 0.001$; Table 2). The PCA clearly separated the bacterial community in the rhizosphere from the bulk soil (Fig. 4C). The effect of the rhizosphere on the bacterial species and functional diversity, and on the bacterial community structure was not affected by the applied agricultural practices ($F = 0.497$, df = 1, $P = 0.617$).

The frequency of 23 ASVs was significantly different between the rhizosphere and the bulk soil, with 17 of them more abundant in the rhizosphere and belonged mainly to Rhizobiales and Burkholderiales and six in the bulk soil belonging mostly to *Halomonas* and *Bacillus* ($P < 0.05$) (Fig. 4D). The frequency of 172 EC numbers was significantly different between the rhizosphere and bulk soil with 41 of them more abundant in the rhizosphere and 131 in the bulk soil (Fig. 3). Functions stimulated in the rhizosphere were related mainly with degradation/utilization/assimilation and to a lesser extent with biosynthesis (Fig. S2). Functions more abundant in the bulk soil were related to biosynthesis and to a lesser extent with degradation/utilization/assimilation, and generation of precursor metabolites and energy.

**Effect of plant development on the bacterial community.** The maize rhizosphere affected the soil bacterial community, but the extent depended on the development stage of the maize plant and on the community trait (Fig. 5A). For instance, the effective number

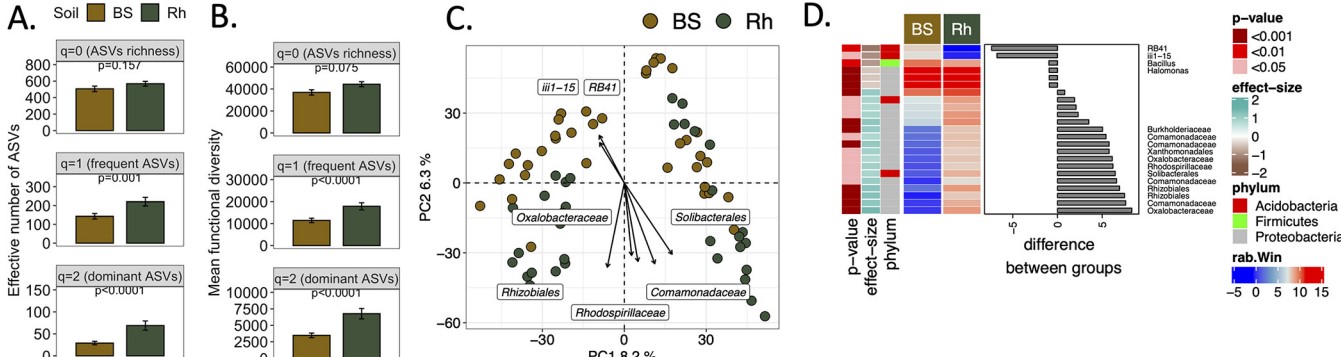

**FIG 4** Effect of maize (*Zea mays* L.) rhizosphere (Rh) versus the bulk soil (BS) on soil bacterial communities. A) Hill numbers of alpha taxonomic diversity, B) Hill numbers of mean functional diversity. The bar height indicates the mean for each type of soil and the error line shows the standard error of the mean. Linear mixed effects models with 1000 Monte-Carlo permutations and plot in the field as random factor was used to test the significant differences (*P*-value). C) Principal-component analysis (PCA) with all amplicon sequence variants (ASVs). D) Differentially abundant ASVs between BS and Rh as determined by an ANOVA-Like Differential Expression tool for compositional data and Benjamini-Hochberg sequential correction. Median centered log-ratio (*clr*) transformed frequencies (rab.win), *P*-values and the effect size were plotted as heat-map and the median difference between species as bar-plot. E) Permutational Analysis of Variance (perMANOVA) with the Aitchinson distances (*n* = 999) to test the main effects, i.e., agricultural practices (treatment), maize developmental stage and their interactions in BS and in the Rh.

of frequent and dominant ASVs was significantly different between the bulk soil and the rhizosphere in the flowering and grain filling stage (*P* < 0.001), but not in the vegetative stage (*P* = 0.169 and *P* = 0.252, respectively; Fig. S3). A similar result was obtained considering the functional diversity, but the rhizosphere had no effect on the species richness in each of the plant development stages.

The bacterial community structure in the bulk soil and rhizosphere was significantly different in the different stages of maize development (*P* < 0.001, Table 2) and the PCA separated the bacterial communities in the different growth stages (Fig. 5B). The effect of the rhizosphere on the bacterial community structure increased over time, i.e., grain filling (*F* = 107.97) > flowering (*F* = 98.98) and > vegetative (*F* = 13.51) (Fig. 5C). The effect, however, was higher in the rhizosphere (*F* = 50.20, *P* < 0.001) than in the bulk soil (*F* = 14.56, *P* < 0.001), i.e., the variation of the bacterial communities during plant growth (rhizosphere) was greater than the temporal variation (bulk soil) (Fig. 4E). The separation between the bulk soil and the rhizosphere was more accentuated in CP than in CA and in the vegetative stage than in the flowering and grain filling stage (Fig. 5D). Consequently, the interaction between the development phases of the maize plant and the agricultural practices had a highly significant effect on the bacterial community structure (*F* = 26.09, df = 2, *P* < 0.001) and was larger in the bulk soil (*F* = 17.07, *P* < 0.001) than in the rhizosphere (*F* = 9.7, *P* < 0.001) (Fig. 4E).

The more abundant ASVs in soil during the vegetative stage belonged to *Stenotrophomonas*, *Bacillus cellulosilyticus*, *Acinetobacter*, *Pseudomonas*, *Halomonas nitritophilus*, *Halomonas*, *Exiguobacterium,* and Xanthomonadaceae (Fig. 5E). The relative frequency of ASVs belonging to *Pseudomonas*, *Halomonas nitritophilus,* and other members of *Halomonas* was significantly higher in the flowering than in the grain filling stage, while ASVs belonging to Bradyrhizobiaceae and *Bradyrhizobium cellulosilyticus* were more abundant in the grain filling than in the flowering stage. Two ASVs were differentially abundant between the bulk soil and the rhizosphere in the vegetative, 20 in the flowering and 8 in the grain filling stage (Fig. S4).

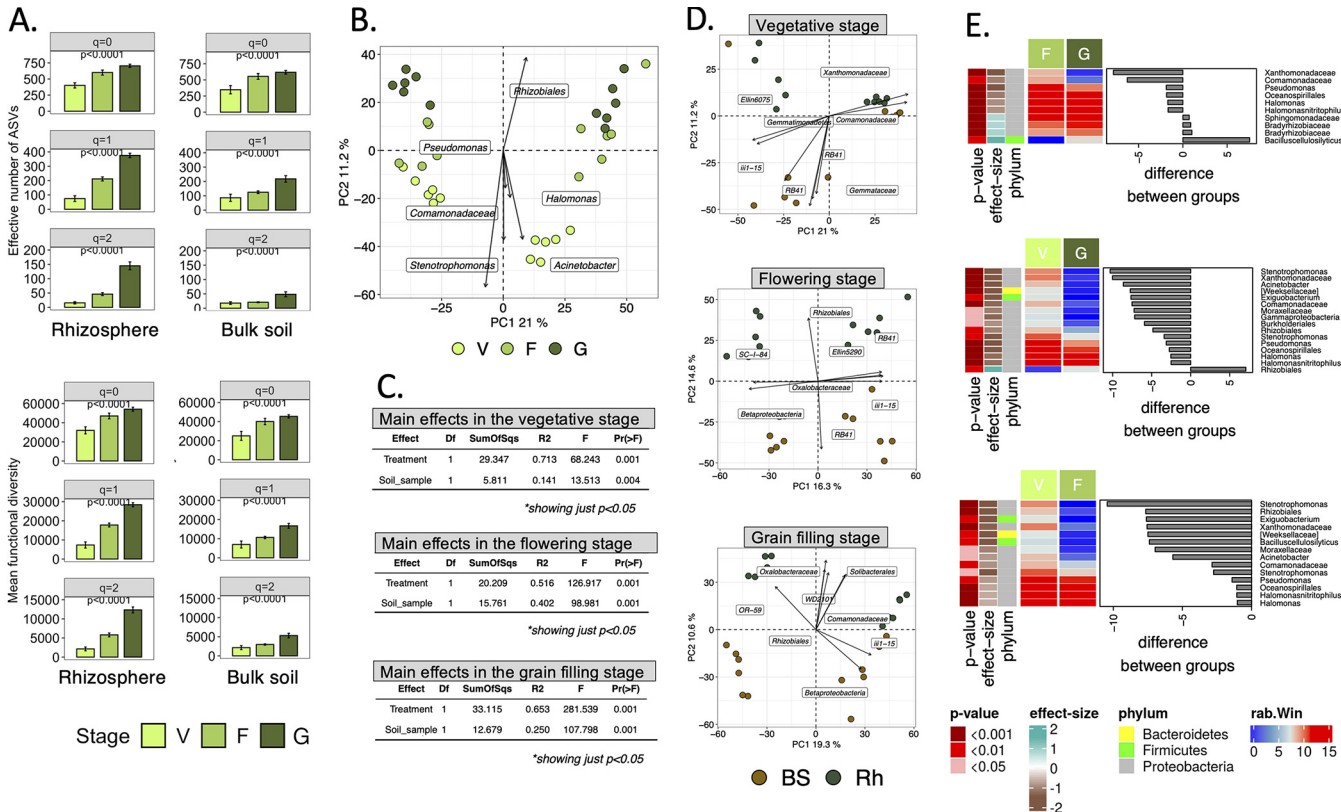

**FIG 5** Effect of maize (*Zea mays* L.) rhizosphere along the vegetative (V), flowering (F) and grain filling (G) growth stages versus the bulk soil (BS) on soil bacterial communities. A) Hill numbers of alpha taxonomic diversity and mean functional diversity. The bar height indicates the mean for each growth stage and the error line shows the standard error of the mean. Linear mixed effects models with 1000 Monte-Carlo permutations and plot in the field as random factor was used to test the significant differences (*P*-value). B) Principal-component analysis (PCA) with all amplicon sequence variants (ASVs) in the rhizosphere soil samples. C) Permutational Analysis of Variance (perMANOVA) with the Aitchinson distances (*n* = 999) to test the main effects, i.e., agricultural practices (treatment) and BS versus in the Rh (soil_sample). D) PCAs with all ASVs in the soil samples at the maize different growth stages. E) Differentially abundant ASVs between V versus F, V versus G and F versus G as determined by an ANOVA-Like Differential Expression tool for compositional data and Benjamini-Hochberg sequential correction. Median centered log-ratio (*clr*) transformed frequencies (rab.win), *P*-values and the effect were plotted as heat-map and the median difference between species as bar-plot.

A total of 227 functions were differentially abundant between the vegetative stage and flowering, and 223 between the vegetative stage and grain filling (Fig. 3) and most of them (80%) were more abundant in the vegetative stage of maize. Functions related to amino acid biosynthesis, cell structure biosynthesis, cofactor, carrier and vitamin biosynthesis, fatty acid and lipid biosynthesis, and nucleoside and nucleotide biosynthesis were enriched in the vegetative stage while cofactor, carrier, and vitamin biosynthesis in flowering, and carbohydrate degradation and terpenoid biosynthesis in the grain filling stage (Fig. S2). One hundred and six functions were differentially abundant between flowering and grain filling, with 87 of them more abundant in the flowering stage than in the grain filling phase (Fig. 3). Functions related to cofactor, carrier, vitamin biosynthesis and amino acid degradation were enriched in the flowering stage while aromatic compound degradation in the grain filling stage (Fig. S2).

**Species turnover between the bulk soil and rhizosphere.** The turnover of total species in the bulk soil and the rhizosphere was significantly higher in soil under CP practices than under CA (Fig. 6). However, the turnover of frequent and dominant species was similar. The turnover of frequent and dominant species in the rhizosphere under CA versus CP was significantly higher (*P* < 0.001) than in the bulk soil, i.e., bacterial communities in the bulk soil of maize resembled more each other even under CA and CP.

## DISCUSSION

After more than 20 years of contrasting management practices, the soil organic C

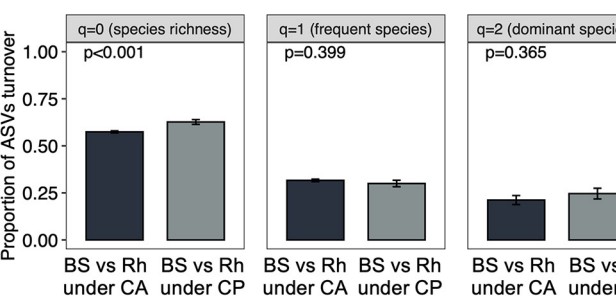 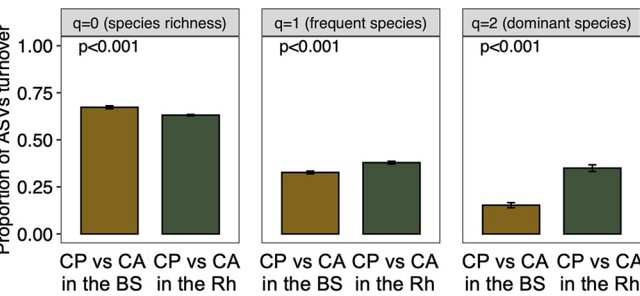

**FIG 6** Hill numbers of the relative amplicon sequence variants (ASVs) turnover rate between soil bacterial communities of the bulk soil (BS) and the maize (*Zea mays* L.) rhizosphere (Rh) under conservational agriculture (CA) or conventional practices (CP), and of the CP versus CA in the BS and CP versus CA in the Rh. The bar height indicates the mean for each comparison and the error line shows the standard error of the mean. Linear mixed effects models with 1000 Monte-Carlo permutations and plot in the field as random factor was used to test the significant differences (*P*-value).

content in CA had increased significantly, 1.3 times, while other soil characteristics, such as pH and EC, remained similar. Soil characteristics, such as pH, clay content and EC, are known to affect the bacterial community structure (23). The pH has often been reported as the major factor that controls the bacterial diversity (24), although changes in EC also have a strong effect on microbial communities (25). As these characteristics, i.e., pH, EC and clay content, were similar in CA and CP, the effect of contrasting management practices on the bacterial community could be investigated.

The amount of organic material applied and how it was managed was different in CA and CP, and that is known to affect the soil bacterial communities (26). In CP only roots and about 10 cm maize stubble were incorporated in soil and in CA all residues of maize and wheat plants were left on the soil surface. We have shown in previous studies at '*El Batán*', that agricultural practices can have a large effect on the bacterial community structure although with large variations over time (27, 28). In this study, a partition analysis of variance revealed that 41.5% of the observed variance was explained by the agricultural practices applied to the system. This factor was the most important in defining the structure of bacterial communities and the effect of agricultural practices was larger than that of the rhizosphere, or plant development. Srour et al. (5) reported that conventional tillage had a larger effect on the soil bacterial community structure than other factors, such as fertilization, in a long-term maize-soybean rotation field experiment as found in numerous other studies (29–32). Although Srour et al. (5) and many others (e.g., [33]), including investigations at '*El Batán*' (16, 27, 28), reported no effect of tillage on diversity and/or species richness, in this new study we found a large negative effect. This observation can be explained, in part, by the method used to calculate diversity. True diversity indices or Hill numbers, which integrate the most common indices to measure diversity in a single framework and quantify diversity in units of equivalent numbers of species or equally abundant species (34), allows a more intuitive interpretation of the diversity concept. The most popular metrics for measuring diversity, such as Shannon or Simpson index, have a certain level of abstraction that are sometimes more difficult to interpret (35). For example, the Shannon index measures entropy, that is, the uncertainty in the species identity of a randomly chosen sequence in the system (36). Soil microbial diversity is considered an indicator of soil health (37) under the premise that ecosystems functions and services are intimately related with biodiversity (38). Using true diversity, we found that CP not only decreased taxonomic diversity, but also functional diversity. Further research is needed to confirm the loss of microbial functions as a result of CP and to investigate its possible effect of on long-term soil fertility.

Tillage in the CP system mixed the roots and stubble with the soil and put it in close direct contact with soil microorganisms, while in the CA soil the crop residue was left on the soil surface. Additionally, tillage breaks up the soil aggregates liberating physically protected soil organic material. Both processes, namely, mixing soil and crop residue, and the release of physically protected organic material from within the

aggregates favor copiotrophs (39). However, tillage occurs occasionally (once after harvest and one to two times during the dry season), so its effect on the microbial community should be limited in time (40). In CA, no disruption of aggregates occurs and the crop residue remains on the soil surface, so it takes longer for the organic material to become available for microorganisms. As such, CA favors oligotrophic microorganisms. The increased abundance of oligotrophs in no-till soils and copiotrophs in tilled soils have been documented in several other studies (5, 16, 27, 30). Ramirez-Villanueva et al. (16) reported that bacterial phyla considered oligotrophs, such as *Acidobacteria*, *Planctomycetes,* and *Verrucomicrobia*, were enriched in CA while bacterial groups considered copiotrophs such as *Actinobacteria*, were enriched in CP. In this study, the oligotrophic phyla *Acidobacteria*, *Planctomycetes,* and *Verrucomicrobia* and *Nitrospira*, and the genera *OR-59* and *DA-101* were enriched in CA compared to CP, but the effect was more outspoken in the rhizosphere than in the bulk soil. In general, more taxonomic groups were enriched in CA, but more putative functions in CP based on the compositional analysis of the functional prediction using 16S rRNA evolutionary trees. Putative functions enriched in CP were mostly related to biosynthesis pathways, while functions such as methanogenesis and $CO_2$ fixation were enriched in CA. This confirms that fast growing copiotrophic bacteria were favored by CP and slow growing k-strategist or oligotrophs by CA.

In the rhizosphere, the increase in easily available organic material due to dying roots and exudates is high and the supply is constant (8) although its composition might vary over time and between plant species (41). The rhizosphere should also stimulate copiotrophs, but the constant supply of organic material might stimulate another type of microorganisms than those enriched by tillage. Our findings supported this hypothesis. The rhizosphere enriched fast growing copiotrophic bacteria, such as Rhizobiales, Sphingomonadales, Xanthomonadales, and Burkholderiales, while in the bulk soil CP enriched other copiotrophs, such as *Halomonas* and *Bacillus*. A larger number of functions related to biosynthesis was enriched in the bulk soil than in the rhizosphere. While functions that responded positively to the exudates in the rhizosphere were mostly related to organic material degradation, mainly of aromatic compounds, carbohydrates and amino acids.

We also hypothesized that the bacterial communities in the rhizosphere of maize cultivated under CA and CP would resemble each other more than in the bulk soil of both treatments as the available organic material in the rhizosphere will be less dependent on agricultural practices and the plant host will define the root microbiome assemblages more than soil characteristics. This hypothesis was rejected. First, a higher turnover of bacterial ASVs occurred in the rhizosphere than in the bulk soil under the two agricultural practices. Second, the interaction of the effect of agricultural practice and soil sample was not statically significant. In other studies, the effect of the rhizosphere on the bacterial groups was affected often by the management practices applied to soil. For instance, a differential response of bacteria in the rhizosphere of maize was observed under different cropping rotation systems (22) and tillage induced specific responses in $N_2$ fixing bacteria in the rhizosphere of wheat (42).

In this study, the maize rhizosphere explained 21% of the total variance of the bacterial community structure. A similar value (29%) was reported by Wattenburger et al. (22). Members of the Rhizobiales, Sphingomonadales, Xanthomonadales, and Burkholderiales, which are known root colonizers were enriched in the rhizosphere of the maize plants in this study. Xu et al. (43) reported *Burkholderia*, *Mesorhizobium,* and *Rhizobium* as core citrus rhizosphere bacteria. Yang et al. (44) reported that members of *Burkholderia*, enriched in the rhizosphere of maize, are known plant growth promoting bacteria due to their capacity to fix $N_2$ and solubilize insoluble phosphates (45). *Sphingobium* is a well-known rhizosphere bacterium (20) and some of its strains have been reported as efficient degraders of aromatic and chlorinated hydrocarbons (46). Putative functions that were also enriched in the rhizosphere of maize plants studied.

The bacterial community structure, and taxonomic and functional diversity of the soil

bacteria in the maize rhizosphere were different during maize development and the differences between the bulk soil and the rhizosphere were more accentuated as the crop developed. Bacterial functions favored in the rhizosphere during plant growth indicated that at the early stages of maize development root exudates stimulated bacterial growth, consequently the negative effect of CP on the bacterial diversity in the rhizosphere was then the most accentuated. By time of flowering, the rhizosphere started to enrich specific bacteria and functions related to cofactors and vitamin synthesis, but functions related to biosynthesis decreased. Finally, at the grain filling stage, the bacterial diversity reached its highest level and functions such as cofactors and vitamin synthesis, and carbohydrate degradation increased. Several studies have reported that the bacterial community composition in the rhizosphere changes during plant development indicating that roots select for different microorganisms at different times (22, 47, 48) most likely through changes in the composition of the root-exudates (21).

The vegetative stage of maize growth is considered one of the most important phases since it signals the transition of the juvenile phase to the adult phase in which a plant becomes competent for sexual reproduction and plays a pivotal role in the subsequent stages of development (49). Conventional practices did affect the bacterial putative functional diversity in the rhizosphere in the vegetative stage, but not later. This early effect of CP on the soil bacterial community might be important for later plant growth.

## CONCLUSION

The agricultural practice applied to the soil was the factor that affected the bacterial community structure most, even more than the maize rhizosphere, plant development stage or temporal variation. Conventional agriculture practices, i.e., tillage and crop residue removal, reduced the taxonomic and functional bacterial diversity in soil compared to conservation agriculture practices, i.e., minimum tillage and crop residue retention. The rhizosphere enriched fast growing copiotrophic bacteria, such as Rhizobiales, Sphingomonadales, Xanthomonadales, and Burkholderiales, while in the bulk soil other copiotrophs were enriched, such as *Halomonas* and *Bacillus*. The bacterial communities in the bulk soil of maize cultivated under CA and CP resembled each other more than they did in the rhizosphere of both treatments. The bacterial community structure, and taxonomic and functional diversity of the soil bacteria in the rhizosphere were different during maize development and the differences between the bulk soil and the rhizosphere were more accentuated as the crop developed. As maize developed, similar bacterial groups were enriched in the rhizosphere. Although an interaction between agricultural practices and rhizosphere on the bacterial structure, and taxonomic and functional diversity was not found, it did in the vegetative stage.

## MATERIALS AND METHODS

**Sampling site and field experiment description.** The '*El Batán*' research station at CIMMYT is located near the former lake Texcoco (Mexico) at 2,249 m.a.s.l. It is situated in the semi-arid, subtropical highlands of Central Mexico (19.318 N; 98.508 W). The slope of the field at the experimental station is <0.3%. The mean maximum and minimum temperatures are 24 and 6°C, respectively (1991–2019) and the average annual rainfall is 637 mm $y^{-1}$, with approximately 551 mm falling between May and October. Short, intense rain showers followed by dry spells typify the summer rainy season and the total yearly potential evapotranspiration of 1,553 mm exceeds annual rainfall. The average growing period at El Batán is 132 days.

The soil at El Batán is classified as a Haplic Phaeozem (Clayic) in the World Reference Base system (50) and as a fine, mixed, thermic Cumulic Haplustoll in the USDA Soil Taxonomy system (51). The soil is characterized by good chemical and physical conditions for farming and the major limitations are periodical drought, periodical water excess and wind and water erosion.

The field experiment is a randomized complete block design with two field replications, i.e., two plots of 7.5 by 22 m for each treatment. The number of replicates used in a field experiment depends largely on the size and cost of the experiment. The practical number of replicates for an experiment is reached when the cost of the experiment is no longer offset by an increase in information gained (52). Experimental designs in agronomy, plant breeding and agriculture, therefore, often use two replicates to reduce costs while still allowing to test for significant differences between treatments (53, 54). Two replicated plots have been used at el Batán since 1991 to reduce cost while providing enough information to determine which agricultural practices, i.e., conventional practices versus conservation

agriculture, improve yields and how they affect soil characteristics, greenhouse gas emissions and microbial communities (e.g., [15, 16, 27, 28, 55]).

There are 32 treatments in the field experiment at El Batán, but for this study only two contrasting treatments were selected: (i) the CA treatment has a rotation of maize and wheat (*Triticum aestivum* L.), zero tillage and retention of all crop residues at the soil surface; (ii) the CP treatment includes monoculture of maize, conventional tillage and removal of crop residues, so the only organic matter incorporated through tillage is about 15 cm of maize stubble and the root biomass. Agronomic management is as follows (56): appropriate available herbicides were used to control weeds as needed, and no disease or insect pest controls were utilized. Maize was planted at 60,000 plants/ha in rows 75 cm apart and wheat in 20-cm rows at 100 kg seed/ha. The planting of both maize and wheat depended on the onset of summer rains but was usually done between 5 and 15 June. Both crops were fertilized at the rate of 120 kg N/ha with urea, with all N applied to wheat before planting and to maize at planting. In the CP system, the tillage operations after harvest consisted of one pass with a chisel plow to 30 cm depth, followed by two passes with a disk harrow to 20 cm depth and two passes with a spring tooth harrow to 10 cm. The spring tooth harrow was used when needed for weed control (typically twice) during the winter fallow season. To prepare the seed bed in May, the tillage operations of December were repeated. Irrigation is applied only when the viability of plants is compromised.

**Soil sampling and physicochemical characteristics.** Soil samples were collected in the crop cycle of 2014. The rhizosphere soil was collected when the maize plants were in the vegetative, flowering and grain filling stage. Within each of the two plots per treatment (CP and CA), three plants were randomly selected to obtain the rhizosphere, at the same time, three samples of the 0–20 cm bulk soil were randomly taken from the center between maize rows (37.5 cm distance to each maize row), so a nested block experimental design was applied ($n = 6$; $k = 12$; $N = 72$; Fig. S1). The field-based replications were maintained for the extraction of DNA and soil characterization.

The rhizosphere was collected from maize plants following the method described by Chaparro et al. (21). The roots were separated from the maize plants and were shaken by hand for 2 min and the soil that did not remain fixed to the roots was discarded. The remaining soil fixed to the roots, i.e., the rhizosphere soil, was collected by shaking the roots in 1 L sterile 0.9% NaCl solution for 10 min. The soil suspension was centrifuged at 150 *g* at room temperature for 15 min. The soil pellet was processed for DNA isolation as described below.

The soil samples were analyzed for particle size distribution by the hydrometer method as described by Gee et al. (57). Total organic carbon (TOC) was measured with a total organic carbon analyzer TOC-VCSN (Shimadzu, Canby, USA) and total nitrogen (N) by the Kjeldahl method using concentrated $H_2SO_4$, $K_2SO_4$, and HgO to digest the sample (58). The water holding capacity (WHC) was determined by subtracting the weight of a given amount of soil saturated with water and filtered overnight minus the weight of dry soil sample. Electrolytic conductivity (EC) was measured in a saturated solution extract as described by Rhoades et al. (59) and pH was measured in 1:2.5 soil-$H_2O$ suspension using a glass electrode (60).

**DNA extraction and PCR amplification of fragment 16S rRNA.** Nine sub-samples of 0.5 g from each soil sample ($N = 72$) were used to extract DNA as described previously by Ramírez-Villanueva et al. (16). As such, 4.5 g soil of each soil sample was extracted for DNA. The PCR amplification of the regions V3-V4 of the 16S rRNA gene from the total DNA were done with primers 341F (5′-CCTACGGGN GGCWGCAG-3′) and 805R (5′-GACTACHVGGGTATCTAATCC-3′) reviewed by Klindworth et al. (61) and containing the adapters for Illumina MiSeq sequencing. The PCR mixture (25 $\mu$l) contained 1 $\times$ reaction buffer, 10 mM each of the four deoxynucleoside triphosphates, 10 $\mu$M with each of the primers, 1.0 U *Taq* polymerase (Thermo Scientific, Waltham, MA, USA), and 5 ng total DNA as the template. The following thermal cycling was used: initial denaturation at 95℃ for 3 min, 25 cycles of denaturation at 95℃ for 30 s, annealing at 55℃ for 30 s and extension at 72℃ for 30 s, followed by a final extension period at 72℃ for 5 min. The PCRs were done in triplicate for each sample, pooled in equimolar amounts, and purified using the QIAquick PCR & Gel Cleanup kit as recommended by the manufacturer (Qiagen, Germany). Quantification of the PCR products was done using PicoGreen dsDNA assay (Invitrogen, Carlsbad, USA) and a nanodrop3300 fluoroespectrometer (Thermo Scientific). Sequencing of the 16S rRNA gene was done using a paired-end 300 pb in an Illumina MiSeq platform by Macrogen Inc. (DNA Sequencing Service, Seoul, South Korea).

**Analysis of the soil microbial community.** Raw sequences were imported into qiime 2–2020.8 and analyzed within QIIME2 (62). Demultiplexing was done with the demux plugin. Denoising, quality filtering, trimming, ASVs dereplication and chimera filtering were done with DADA2 (63). The filter and trim parameters were truncLen = 120, trimLeft = 0, maxEE = 2, and truncQ = 2. Query sequences, from now on feature data, were taxonomy assigned with classify-sklearn with a Naive Bayes supervised learning algorithm using the trained Greengenes reference database *gg-13-8-99-515-806-nb*. An alignment filtering method was applied to the feature data using *vsearch* with the parameters perc-identity = 0.97 and perc-query-aligned = 0.95. Sequences that did not align were deleted and also organellar 16S rRNA sequences, i.e., from mitochondria and chloroplast.

The functional genes were predicted with PICRUSt2 (https://github.com/picrust/picrust2; v2.0.0-b). First, the feature data and a reference database of genomes from the Integrated Microbial Genomes database were aligned with hidden Markov models to insert the ASVs into a reference tree. The genome prediction was done with a hidden-state algorithm. Pathway abundances based on EC number abundances were inferred with MetaCyc (64).

**Statistical analysis.** The table with frequencies of each ASV, from now on feature table, taxonomy data, EC numbers table of frequencies, and metadata were imported into R for downstream analyses

with the R package mentioned below (65). The alpha taxonomic diversity was calculated with the feature table and beta diversity with Hill-numbers-based similarity measures (66–68). The relative species turnover rate per community, which represents the proportion of a typical community that changes from one to another community, was quantified with the Metagenome Diversity script in R (69). First, multiplicative partition of beta diversity was calculated with:

$$_qD_ß = {_qD_\gamma}\big/{_qD_\alpha} \qquad \text{(Eq. 1)}$$

where $_qD_ß$ is beta diversity at $q$ order, $_qD_\gamma$ is gamma diversity at $q$ order and $_qD_\alpha$ is alpha diversity at $q$ order.

Equations to determine alpha and gamma diversity can be found in Ma and Li (69). The relative species turnover rate per metagenome $V_{qN}$ was calculated with:

$$V_{qN} = {_q}\frac{D_ß - 1}{N - 1} \qquad \text{(Eq. 2)}$$

where N is the number of metagenome samples. This parameter is equivalent to the Sørensen dissimilarity measure when $q = 0$ and equivalent to the Morisita-Horn dissimilarity index when $q = 2$. Functional diversity was determined with the mean functional diversity per species (MD_q), which calculates the effective sum of pairwise distances between a fixed species and all other species, using the feature table (community) and the EC numbers table of frequencies (traits) with the HillR (v.0.5.1) package (70).

The effect of agricultural practices (CA versus CP), the cultivation of maize (bulk soil versus the rhizosphere) and growth stage of maize (vegetative versus flowering versus grain filling) on diversity parameters was determined with a linear mixed effects model (LMEM) using the field plot as random factor with the nlme (v.3.1-151) package (71). The significance of the effect was calculated with 1000 Monte Carlo samplings using the pgirmess (v.1.7.0) package (72).

A compositional approach was used to investigate the community composition and analyze the structure (73). First, the zero count values were replaced on the feature table and EC numbers table of frequencies with a count zero multiplicative (CZM) method using the 'cmultRepl' function in zCompositions (v.1.3.4) package (74). Data sets were centered log-ratio (clr) transformed with the 'codaSeq.clr' function in CoDaSeq package (75). Distance pairwise matrices were calculated using the Aitchison distance (73). The perMANOVA ($n = 999$) was done on Aitchison distance matrices to test the main effects, i.e., agricultural practices, rhizosphere and growth stage of maize, and the random effects, i.e., the plots in the field, and their interactions using vegan (v.2.5-7) package (76). As the plots in the field had no significant effect on composition ($R^2 = 0.012$, $P = 0.08$), they were not included in downstream analyses. Subsets with the samples of bulk soil or rhizosphere were taken to test the effect of agricultural practices and growth stage of maize, and subsets of each growth stage to test the effect of agricultural practices and the maize rhizosphere with a perMANOVA analysis. The PCAs were done with the 'prcomp' function in stats (v.4.0.5) package (65) using the clr transformed data. The covariance structure of the compositional data are highly biased, and the results of a multivariate analysis are not reliable without a suitable transformation of the data to visualize a possible effect of agricultural practices, rhizosphere and growth stage of maize on soil bacterial communities. The ALDEx analysis was done using ALDEx2 (v.1.18) package (77, 78). Raw counts were used as input and Monte Carlo Dirichlet instances of the clr transformation values were generated with the function 'aldex.clr'. Test for differentially abundant taxa or functions was done with the function 'aldex.glm' that calculates the expected values for each coefficient of a general lineal model (glm). Benjamini-Hochberg sequential correction was applied to the resulting $P$-value to avoid inflation of Type-I error. Heatmaps of the differentially abundant taxa and functions were constructed with the ComplexHeatmap (v.2.6.2) package (79).

**Data availability.** Raw sequences can be found in the Sequence Read Archive (SRA) of the NCBI with the BioProject accession number PRJNA436368. Scripts and R programing codes can be found in github (https://github.com/steph0522/Maize_rhizosphere_paper.pdf).

## SUPPLEMENTAL MATERIAL

Supplemental material is available online only.
**SUPPLEMENTAL FILE 1**, PDF file, 1.3 MB.

## ACKNOWLEDGMENTS

This research was funded by 'Centro de Investigación y de Estudios Avanzados del IPN' (CINVESTAV), 'Fondo Sectorial de Investigación para la educación SEP-CONACYT Convocatoria CB-2015-01 (Project Number, 252080),' 'Apoyo Especial para Fortalecimiento de Doctorado PNPC 2013,' and project 'Infraestructura 205945' from 'Consejo Nacional de Ciencia y Tecnología' (CONACyT, Mexico). The long-term experiment forms part of the strategic research for 'Cultivos para México/MasAgro' and CRP MAIZE, supported by SADER and various W1 and W1 donors. Y.C.-R., A.S.D.-L., and S.H.-P. received a grant-aided from CONACyT.

Y.E.N.-N. and L.D. conceived the study, analyzed data, interpreted the results, and wrote the paper; Y.C.-R. and A.S.D.-L. did the experimental laboratory work; S.H.-P.

analyzed the data; N.V. and B.G. maintained the field experiment, conceived the study, and revised the manuscript.

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
