## [Reviewer comments · Microbiology Spectrum]

Microbiology Spectrum

Bacterial communities in the rhizosphere at different growth stages of maize cultivated in soil under conventional and conservation agricultural practices

Yendi Navarro-Noya, Stephanie Hereira-Pacheco, Yosef Chávez-Romero, Bram Govaerts, Nele Verhults, Arit De León-Lorenzana, and Luc Dendooven

Corresponding Author(s): Yendi Navarro-Noya, Universidad Autónoma de Tlaxcala

Review Timeline:

Submission Date:	October 7, 2021
Editorial Decision:	December 9, 2021
Revision Received:	February 2, 2022
Accepted:	February 4, 2022

Editor: Kristen DeAngelis

Reviewer(s): The reviewers have opted to remain anonymous.

Transaction Report:

DOI: <https://doi.org/10.1128/spectrum.01834-21>

December 9, 2021

Dr. Yendi E Navarro-Noya
Universidad Autónoma de Tlaxcala
Centro de Investigación en Ciencias Biológicas
Tlaxcala, Tlaxcala 90000
Mexico

Re: Spectrum01834-21 (Bacterial communities in the rhizosphere at different growth stages of maize cultivated in soil under conventional and conservation agricultural practices)

Dear Dr. Yendi E Navarro-Noya:

Thank you for submitting your manuscript to Microbiology Spectrum. Both reviewers felt that while this study was conducted with integrity and presented well, that there were some deficiencies that need to be addressed. While I am passing along the reviewer comments in their entirety, I wish to point out that novelty of findings are specifically not required for publication in Microbiology Spectrum. I also recognize that repeating a field experiment might not be feasible, but a stronger defense of the findings in the context of prior research could make up for this limitation.

Link Not Available

Sincerely,

Kristen DeAngelis

Journals Department
Reviewer comments:

Reviewer #1 (Comments for the Author):

The manuscript presents an in-depth evaluation of the microbial community associated with maize fields in central Mexico. Fields cultivated under conventional practices (CP) and conservation agriculture (CA) were examined. Both bulk and rhizosphere samples, as well as samples obtained at different maize growth stages were assessed. The impact of all such factors on the microbial diversity, community structure, and functional potential is reported. Few hypotheses (e.g., effect of agricultural practice on proportion of copiotrophs versus oligotrophs) were tested. Special attention was given to deciphering the relative contribution of each of the factors described above to the observed changes in the community. The general conclusion is that CA positively impacted soil quality (organic matter content), diversity, and induced shifts in community structure.

General comment. The manuscript presents a thorough and well written diversity survey of an interesting and economically

relevant ecosystem. It adds to the current state of knowledge regarding the impact of agricultural practices on the soil microbial community. The discussion successfully put the results in context of prior studies. The experimental design is sound and the experimental and analysis aspects of the manuscript appear to be conducted with care.

Specific comments.

1. In the abstract, please clarify the methods used. Add a sentence like "16S rRNA high throughput diversity survey using the Illumina platform was used to assess diversity and community structure, and functional diversity was inferred from taxa identity using PieCrust. This is especially important for the functional part. It is not clear to the reader how it was examined, till you get to the end of the methods section. Also, please clarify that these are predicted or inferred functional predictions, not true ones that was obtained by metagenomic or metatranscriptomic analysis. The authors refer to this approach in L104 as "ancestral reconstruction approach". I am not sure, how this method could be described as such.
2. Results L107, and at the start of various sections. Since the methods comes after the results in this journal, it would be helpful to add a sentence at the start of each section in the results to prime the reader to what's being presented. For example, at L107 a sentence stating that x samples from y treatments with z replicates were obtained, and the physicochemical properties were determined (see methods). Same with other sections e.g. at L206.
3. How was species turnover calculated. The methods (L456) state that "was quantified with the MetagenomeDiversity script in R". Could you please provide more details on the scientific/mathematical basis of this calculation

Reviewer #2 (Comments for the Author):

General and Specific Comments:

The purpose of this study was to determine the effect of agricultural practices and the maize rhizosphere on soil bacterial communities. Bulk and maize rhizosphere soil under conventional management practices (CP) and conservation agricultural practices (minimum tillage, crop residue retention and crop diversification) (CA) were sampled during the vegetative, flowering and grain filling stage, and the taxonomic functional diversity, and community structure of the bacterial community were determined. Overall, this is a well-planned and well-executed study, however the study has several deficiencies.

- 1) All studies, including field studies, need to be repeated at least once. This study was conducted only one time in 2014.
- 2) Microbiome studies like this typically use three independent replications. However, it appears that the field plot was established with only 2 replicates per treatment. This deficiency would be of less concern if the study was repeated.
- 3) The results of this study lack novelty and demonstrate what is already well described in the microbiome literature. For example: a) analysis indicated a large and highly significant effect of agricultural practice on the bacterial community; b) the rhizosphere had a highly significant effect on the bacterial community structure; c) the community structure differed between the rhizosphere and bulk soil; d) the bacterial community structure and taxonomic and functional diversity of the soil bacteria in the maize rhizosphere were different during maize development.
- 4) The conclusion that agricultural practice was the factor that affected the bacterial community the most, is not surprising. The CA was a maize/wheat rotation and CP was continuous maize. It would be expected that cropping wheat every other year would have a profound effect on the soil microbiome. To determine the effect of management practice (for example: conventional tillage vs. reduced tillage), continuous maize should have been grown in both treatments.
- 5) A brief description of the agronomic management practices of both treatments should be mentioned even though it is described in reference 56.
- 6) More details are needed about how the soil and rhizosphere samples were collected and processed.
- 6) Please highlight how this study moves our understanding of the microbiome of cereal crops forward?

Staff Comments:

Preparing Revision Guidelines

Please return the manuscript within 60 days; if you cannot complete the modification within this time period, please contact me. If you do not wish to modify the manuscript and prefer to submit it to another journal, please notify me of your decision immediately so that the manuscript may be formally withdrawn from consideration by Microbiology Spectrum.

Reviewer comments:

Reviewer #1 (Comments for the Author):

The manuscript presents an in-depth evaluation of the microbial community associated with maize fields in central Mexico. Fields cultivated under conventional practices (CP) and conservation agriculture (CA) were examined. Both bulk and rhizosphere samples, as well as samples obtained at different maize growth stages were assessed. The impact of all such factors on the microbial diversity, community structure, and functional potential is reported. Few hypothesis (e.g., effect of agricultural practice on proportion of copiotrophs versus oligotrophs) were tested. Special attention was given to deciphering the relative contribution of each of the factors described above to the observed changes in the community. The general conclusion is that CA positively impacted soil quality (organic matter content), diversity, and induced shifts in community structure.

General comment. The manuscript presents a thorough and well written diversity survey of an interesting and economically relevant ecosystem. It adds to the current state of knowledge regarding the impact of agricultural practices on the soil microbial community. The discussion successfully put the results in context of prior studies. The experimental design is sound and the experimental and analysis aspects of the manuscript appear to be conducted with care.

We thank the reviewer for her/his positive comments on our investigation.

Specific comments.

1. In the abstract, please clarify the methods used. Add a sentence like "16S rRNA high throughput diversity survey using the Illumina platform was used to assess diversity and community structure, and functional diversity was inferred from taxa identity using PieCrust. This is especially important for the functional part. It is not clear to the reader how it was examined, till you get to the end of the methods section. Also, please clarify that these are predicted or inferred functional predictions, not true ones that was obtained by metagenomic or metatranscriptomic analysis.

We changed the abstract to avoid confusion as the reviewer suggested.

“Bulk and maize (*Zea mays* L.) rhizosphere soil under conventional practices (CP) and CA were sampled during the vegetative, flowering and grain filling stage, and 16S rRNA metabarcoding was used to assess bacterial diversity and community structure. The functional diversity was inferred from the bacterial taxa using PICRUSt.”

The authors refer to this approach in LI04 as "ancestral reconstruction approach". I am not sure, how this method could be described as such.

We used the description mentioned in the PICRUSt paper (Langille et al. 2013) “PICRUSt uses an extended ancestral-state reconstruction algorithm to predict which gene families are present and then combines gene families to estimate the composite metagenome”.

However, to avoid confusion we changed the text.

“The functional profile of the bacteriome was obtained with the “Phylogenetic Investigation of Communities by Reconstruction of Unobserved States” software (PICRUST). The bacterial alpha and beta, and functional diversity was determined.”

2. Results L107, and at the start of various sections. Since the methods comes after the results in this journal, it would be helpful to add a sentence at the start of each section in the results to prime the reader to what's being presented. For example, at L107 a sentence stating that x samples from y treatments with z replicates were obtained, and the physicochemical properties were determined (see methods). Same with other sections e.g. at L206.

The reviewer is right, and we missed this important point. We tried to add that information to the Results section as suggested by the reviewer.

“A total of 72 soil samples were collected ($N = 72$), i.e. two agricultural practices (CP and CA) \times two different soil samples (bulk and rhizosphere soil) \times collected three times during maize plant development (vegetative, flowering and grain filling stage) \times from two plots and \times three maize plants in each plot, and the physicochemical characteristics were determined (see Material and methods).”

“Metabarcoding analysis was done with the V3-V4 regions of the 16S rRNA genes and sequenced with 300-PE runs in an Illumina MiSeq. Sequencing retrieved a total of 4,324,922 high quality sequences representing 4,546 amplicon sequence variants (ASVs).”

“A compositional approach was used to investigate the bacterial community composition and analyze its structure (see Materials and methods; 23).”

“The principal component analysis (PCA) clearly separated the bacterial communities in CA from those in CP (Fig. 2C). The permuted multivariate analysis of variance (perMANOVA) with Aitchison distances to test the effect of the agricultural practices indicated...”

“The effect of different treatments on the ASVs and enzyme classification numbers (EC numbers) were determined with an ANOVA-like differential expression (ALDEx; 24).”

3. How was species turnover calculated. The methods (L456) state that "was quantified with the MetagenomeDiversity script in R". Could you please provide more details on the scientific/mathematical basis of this calculation.

We included the information requested by the reviewer.

“First, multiplicative partition of beta diversity was calculated with:

$${}_qD_\beta = {}_qD_\gamma / {}_qD_\alpha \quad (\text{Eq. 1})$$

where ${}_qD_\beta$ is beta diversity at order q , ${}_qD_\gamma$ is gamma diversity at order q and ${}_qD_\alpha$ is alpha diversity at order q .

Equations to determine alpha and gamma diversity can be found in Ma and Li (71).

The relative gene turnover rate per metagenome V_{qN} was calculated with:

$$V_{qN} = \frac{qD_{\beta} - 1}{N - 1} \quad (\text{Eq. 2})$$

where N is the number of metagenome samples.

This parameter is equivalent to the Sørensen dissimilarity measure when $q = 0$ and equivalent to the Morisita-Horn dissimilarity index when $q = 2$.”

Reviewer #2 (Comments for the Author):

General and Specific Comments:

The purpose of this study was to determine the effect of agricultural practices and the maize rhizosphere on soil bacterial communities. Bulk and maize rhizosphere soil under conventional management practices (CP) and conservation agricultural practices (minimum tillage, crop residue retention and crop diversification) (CA) were sampled during the vegetative, flowering and grain filling stage, and the taxonomic functional diversity, and community structure of the bacterial community were determined. Overall, this is a well-planned and well-executed study, however the study has several deficiencies.

1) All studies, including field studies, need to be repeated at least once. This study was conducted only one time in 2014.

We agree with the reviewer that studies mostly related to crop production are normally done on replicated plots ($n \geq 3$) to account for spatial variability and repeated over three growing seasons so that the effect of climatical conditions are accounted for. In this study, however, we wanted to know how a developing rhizosphere (vegetative, flowering and grain filling) within specific conditions (moisture and temperature) affected the bacterial community structure. We included spatial and plant variability (three plants per plot). A measurement over more than one crop season would add an additional variable (climatical conditions) to our investigation, i.e. the semiarid highlands of Mexico are characterized by variable precipitation that affects plant growth (the plots are not irrigated), the rhizosphere microorganisms and the interaction between the plants and the microorganisms in the rhizosphere. However, this was not the objective of this study and including different growing seasons would make our study even more ambitious, i.e. we would add one factor more. We think that our study is already wide ranging considering the different factors we studied which obliges us to reduce the data that we can report. The study suggested by the reviewer is very interesting and we would like to study the effect of variation in climatical conditions (growing seasons) on the bacterial community, but we would not include temporal variability over one growing season as this would overload the experiment.

2) Microbiome studies like this typically use three independent replications. However, it appears that the field plot was established with only 2 replicates per treatment. This deficiency would be of less concern if the study was repeated.

We agree with the reviewer that 2 replicated plots might reduce the possibility to determine clear effects on the bacterial community structure. CIMMYT decided to include two replicated plots at El Batán to reduce the costs of its field experiments. The variation in managing the field experiments (e.g. sowing, tillage, N fertilizer application, crop residue management) are kept to a minimum so that a possible effect of the agricultural practices studied can be determined. Although these field experiment have only two replicated plots, we have used them in numerous studies and have been able to determine the effect of different agricultural practices or application of organic material on the soil microbiota (e.g. Ceja-Navarro et al., 2010; Navarro-Noya et al. 2013; Ramírez Villanueva et al., 2015; Romero-Salas et al., 2020). We agree with the reviewer that more replicated plots would be better than two plots, but we have found that it is possible to quantify the possible effect of

the agricultural practices on soil microbiota, C and N mineralization and greenhouse gas emissions.

3) The results of this study lack novelty and demonstrate what is already well described in the microbiome literature. For example: a) analysis indicated a large and highly significant effect of agricultural practice on the bacterial community; b) the rhizosphere had a highly significant effect on the bacterial community structure; c) the community structure differed between the rhizosphere and bulk soil; d) the bacterial community structure and taxonomic and functional diversity of the soil bacteria in the maize rhizosphere were different during maize development.

As far as we know very few investigations have been published that study the effect of agricultural practices on the bacterial community at three stages of maize development in semi-arid regions of the world. We stated in the introduction that different patterns emerged depending on the ecosystem, and the effect of the different factors studied (agricultural practices, maize rhizosphere, and plant stage) are not so straightforward, but depend on many other factors (e.g. climate, type of soil) that may be related to the environment in which the agroecosystem is found. The experimental field is in a semi-arid environment which predominates in Mexico and in the world. Our research provides evidence that in the environment studied with that crop in that type of soil, agricultural practices have a very important effect, not only on the bacterial community structure, but also on bacterial diversity. The latter is usually highly variable across different studies, although it might be as diversity metrics are sometimes misinterpreted. Another observed effect was that there is no interaction between agricultural practices and the effect of maize on the soil microbiota, i.e. the effect of the plant is the same regardless of the agricultural practice applied. We stressed in the Introduction section and in the Abstract importance that the experimental field is located in a semi-arid environment.

4) The conclusion that agricultural practice was the factor that affected the bacterial community the most, is not surprising. The CA was a maize/wheat rotation and CP was continuous maize. It would be expected that cropping wheat every other year would have a profound effect on the soil microbiome. To determine the effect of management practice (for example: conventional tillage vs. reduced tillage), continuous maize should have been grown in both treatments.

The experiment at CIMMYT was started to study the possible effect of conventional agricultural practices, i.e. maize monoculture, tillage and crop residue removal, versus more sustainable conservation agriculture, i.e. crop rotation, minimum tillage and crop residue retention. We wanted to investigate how these two contrasting agricultural practices affected the bacterial community in the rhizosphere, i.e. what is the effect of intensive agriculture practices versus that of a more sustainable system. We know already that there is a profound effect of agricultural practices on soil characteristics, GHG emissions and C and N dynamics so we wanted to investigate “how do they affect the rhizosphere bacterial community”.

In previous studies, we found that maize-wheat crop rotation vs maize monoculture had no effect on the soil microbiota (e.g. Navarro-Noya et al. 2013). We hypothesized that since

both wheat and maize are Gramineae, the effect is much smaller than if the rotation included leguminous plants (maize-bean rotation).

Additionally, a wheat monoculture was not included in the experimental set-up at el Batán as it is not a common agricultural practice in Mexico, but maize monoculture is still widely used in conventional intensive agriculture.

We changed the manuscript in the parts where we referred only to the tillage and replaced with CP instead.

“Using true diversity, we found that ~~tillage and crop residue removal~~ CP not only decreased taxonomic diversity, but also functional diversity. Further research is needed to confirm the loss of microbial functions as a result of ~~intensive tillage~~ CP and to investigate its possible effect of on long-term soil fertility.”

“Bacterial functions favored in the rhizosphere during plant growth indicated that at the early stages of maize development root exudates stimulated bacterial growth, consequently the negative effect of ~~intensive tillage~~ CP on the bacterial diversity in the rhizosphere was then the most accentuated.”

5) A brief description of the agronomic management practices of both treatments should be mentioned even though it is described in reference 56.

We included the information in the Materials and methods section.

“Agronomic management is as follows (58): appropriate available herbicides were used to control weeds as needed, and no disease or insect pest controls were utilized. Maize was planted at 60,000 plants/ha in rows 75 cm apart and wheat in 20-cm rows at 100 kg seed/ha. The planting of both maize and wheat depended on the onset of summer rains but was usually done between 5 and 15 June. Both crops were fertilized at the rate of 120 kg N/ha with urea, with all N applied to wheat before planting and to maize at planting. In the CP system, the tillage operations after harvest consisted of one pass with a chisel plough to 30 cm depth, followed by two passes with a disk harrow to 20 cm depth and two passes with a spring tooth harrow to 10 cm. The spring tooth harrow was used when needed for weed control (typically twice) during the winter fallow season. To prepare the seed bed in May, the tillage operations of December were repeated. Irrigation is applied only when the viability of plants is compromised”

6) More details are needed about how the soil and rhizosphere samples were collected and processed.

We included a more detailed description of how the soil and rhizosphere samples were collected and processed.

“The rhizosphere was collected from maize plants following the method described by Chaparro et al. (21). The roots were separated from the maize plants and were shaken by hand for 2 min and the soil that did not remain fixed to the roots was discarded. The remaining soil fixed to the roots, i.e. the rhizosphere soil, was collected by shaking the roots in 1 L sterile 0.9 % NaCl solution for 10 min. The soil suspension was centrifuged at 150 g at room temperature for 15 min. The soil pellet was processed for DNA isolation as described below”

6) Please highlight how this study moves our understanding of the microbiome of cereal crops forward?

The most important conclusion for us is that we found that under the given experimental conditions (climate, agricultural practices, crop), the soil bacterial diversity is higher under conservation agriculture than in conventional agricultural practices, including the rhizosphere of plants. This might have an important effect on nutrient cycling, pathogens and the plant microbiome.

We added this conclusion to the text so that as suggested by the reviewer.

February 4, 2022

Dr. Yendi E Navarro-Noya
Universidad Autónoma de Tlaxcala
Centro de Investigación en Ciencias Biológicas
Tlaxcala, Tlaxcala 90000
Mexico

Re: Spectrum01834-21R1 (Bacterial communities in the rhizosphere at different growth stages of maize cultivated in soil under conventional and conservation agricultural practices)

Dear Dr. Yendi E Navarro-Noya:

Thank you for your careful attention to reviewer comments. Your manuscript has been accepted, and I am forwarding it to the ASM Journals Department for publication. You will be notified when your proofs are ready to be viewed.

Sincerely,

Kristen DeAngelis
Editor, Microbiology Spectrum

Journals Department
S1-4: Accept